# Estrogen-sensitive medial preoptic area neurons coordinate torpor in mice

Zhi Zhang [1,2], Fernando M. C. V. Reis [3], Yanlin He [4,5], Jae W. Park[1], Johnathon R. DiVittorio[1], Nilla Sivakumar[1], J. Edward van Veen [1,2], Sandra Maesta-Pereira [3], Michael Shum[6,7], India Nichols [1], Megan G. Massa [1,2], Shawn Anderson[1], Ketema Paul[1], Marc Liesa [6,7], Olujimi A. Ajijola [8], Yong Xu [4], Avishek Adhikari [2,3] & Stephanie M. Correa [1,2 ✉]

Homeotherms maintain a stable internal body temperature despite changing environments. During energy deficiency, some species can cease to defend their body temperature and enter a hypothermic and hypometabolic state known as torpor. Recent advances have revealed the medial preoptic area (MPA) as a key site for the regulation of torpor in mice. The MPA is estrogen-sensitive and estrogens also have potent effects on both temperature and metabolism. Here, we demonstrate that estrogen-sensitive neurons in the MPA can coordinate hypothermia and hypometabolism in mice. Selectively activating estrogen-sensitive MPA neurons was sufficient to drive a coordinated depression of metabolic rate and body temperature similar to torpor, as measured by body temperature, physical activity, indirect calorimetry, heart rate, and brain activity. Inducing torpor with a prolonged fast revealed larger and more variable calcium transients from estrogen-sensitive MPA neurons during bouts of hypothermia. Finally, whereas selective ablation of estrogen-sensitive MPA neurons demonstrated that these neurons are required for the full expression of fasting-induced torpor in both female and male mice, their effects on thermoregulation and torpor bout initiation exhibit differences across sex. Together, these findings suggest a role for estrogen-sensitive MPA neurons in directing the thermoregulatory and metabolic responses to energy deficiency.

[1] Department of Integrative Biology and Physiology, University of California Los Angeles, Los Angeles, CA, USA. [2] Brain Research Institute, University of California Los Angeles, Los Angeles, CA, USA. [3] Department of Psychology, University of California Los Angeles, Los Angeles, CA, USA. [4] USDA/ARS Children's Nutrition Research Center, Department of Pediatrics, Baylor College of Medicine, Houston, TX, USA. [5] Pennington Biomedical Research Center, Louisiana State University, Baton Rouge, LA, USA. [6] Division of Endocrinology, Department of Medicine, David Geffen School of Medicine, Los Angeles, CA, USA. [7] Department of Molecular and Medical Pharmacology, David Geffen School of Medicine, Los Angeles, CA, USA. [8] UCLA Cardiac Arrhythmia Center, Department of Medicine, David Geffen School of Medicine, Los Angeles, CA, USA. ✉email: stephaniecorrea@ucla.edu

Maintaining a stable internal body temperature requires complex responses that coordinately modulate facultative thermogenesis, heat dissipation, and basal metabolic rate. These thermoregulatory effectors are energetically demanding and may be downregulated when energy is scarce. Thus, when food is unavailable, some mammalian species can enter a regulated state called torpor, in which body temperature, physical activity, metabolism, and reproduction are dramatically reduced[1–3]. Pioneering studies have identified the preoptic area (POA) as a key regulator of temperature and energy balance[4,5]. There are neuronal populations within the POA that respond to local temperature changes and receive ascending temperature signals from the periphery[6–8]. Integrative temperature information in the POA is then transmitted to various brain sites that modulate physiological and behavioral thermal responses[9,10]. In addition to thermoregulation, the POA also monitors metabolic state and regulates food intake[11]. The dual functions of the POA in maintaining temperature and energy balance suggest the hypothesis that this region may coordinate the thermal and metabolic responses to energy deficiency. Elegant studies have begun to pinpoint neuron populations in the POA and associated neural circuits that control body temperature (reviewed in refs. [9,10,12]); however, it is unclear if these or other neuron populations in the POA orchestrate the thermoregulatory and the metabolic changes observed in torpor.

The medial POA (MPA)[13,14] is densely enriched for estrogen receptor alpha (ERα, gene Esr1). Whereas estrogen signaling in the POA has well-characterized effects on sexual, parental, and aggressive behaviors[15–17], the role of ERα signaling in the POA on thermoregulation and metabolism is less clear. Estrogens are potent modulators of both temperature and metabolism[18,19]. Daily body temperature fluctuates during the menstrual cycle in women[20,21] and the estrous cycle in mice[22,23], and estradiol administration alters body temperature in humans[24,25] and rodents[26,27]. Within the hypothalamus, ERα signaling modulates various metabolic processes, including glucose metabolism[28,29], food intake[30], thermogenesis[31], physical activity[32], and adiposity[33,34]. Single-cell RNA profiling of the POA has revealed that Esr1 transcripts are expressed within many of the neuronal clusters that control body temperature and metabolism[14]. Considering the overlapping functions of the POA and estrogens on thermoregulation and energy homeostasis, we questioned whether ERα positive (ERα+) neurons in the MPA also coordinate torpor. Specifically, we asked if ERα+ MPA neurons are sufficient to reduce body temperature and metabolism as observed in torpor, exhibit changes in endogenous activity during torpor, and are required for the expression of torpor in mice. Here, we find that chemogenetic activation of ERα+ MPA neurons is sufficient to induce hypothermia and hypometabolism, that the natural activity of ERα+ MPA neurons is elevated during fasting-induced torpor, and that ERα+ MPA neurons are required for the full expression of fasting-induced torpor in mice.

## Results

### ERα+ neurons in the MPA drive a rapid decrease in core body temperature.

To visualize the anatomical distribution of ERα neurons in the MPA, we evaluated ERα immunoreactivity in brain sections ranging from anterior POA (Bregma 0.6 mm) to posterior POA (Bregma −0.4 mm) according to the mouse brain atlas[35] in both male and female mice. We found enrichment of ERα+ cells within the anatomical boundaries of the MPA, particularly within the medial preoptic nucleus (MPN) and rostral sections of the MPA (Fig. 1a). Overall, ERα immunoreactivity in the MPA was higher in females than in males (Fig. 1a and Supplementary Fig. 1a, b). These patterns of expression are consistent

with previous findings showing ERα immunoreactivity[13] or Esr1 transcripts[14] in the mouse MPA and sexually dimorphic ERα expression in this and associated hypothalamic regions[36,37]. Additionally, Esr1 is co-expressed with transcripts that mark warm-responsive[14,38] or torpor-regulating neurons[39,40] in the MPA.

To test the role of ERα+ MPA neurons in torpor, we investigated the effect of selectively activating ERα+ MPA neurons using chemogenetic Designer Receptors Exclusively Activated by Designer Drugs (DREADDs), a powerful tool for modulating neuronal activity[41]. An adeno-associated virus (AAV) construct encoding the Cre-dependent, excitatory, Gq-coupled hM3Dq receptor and an mCherry reporter was stereotaxically injected into the MPA of mice that express Cre in Esr1-expressing cells (Esr1Cre) and wild-type littermates (Fig. 1b). mCherry was detected in the MPA of female and male Esr1Cre mice (Fig. 1c and Supplementary Fig. 1c) but not in wild-type controls that received the same AAV. Upon intraperitoneal administration of the exogenous ligand for the hM3Dq receptor, clozapine-N-oxide (CNO, 0.3 mg/kg), male and female Esr1Cre mice exhibited a rapid reduction in core body temperature (Fig. 1d, e and Supplementary Fig. 1d, e). Core temperature dipped below 30 °C within 1 h of CNO injection (Fig. 1e and Supplementary Fig. 1e) but was not altered in the same mice following saline injection on a different day (Fig. 1d, e and Supplementary Fig. 1d, e). The effect on body temperature cannot be attributed to CNO alone or its conversion to clozapine, as CNO did not alter body temperature in wild-type mice that received AAV (Supplementary Fig. 1f). In addition, administration of an alternative DREADD ligand, Compound 21 (1 mg/kg)[42], elicited a reduction in body temperature similar to CNO (Supplementary Fig. 1g).

Thermal homeostasis is maintained by the balance of heat production and heat dissipation. In rodents, the brown adipose tissue (BAT) is critical in adaptive thermogenesis[43,44], whereas a modulation of blood flow to the tail skin plays an active role in heat dissipation[45,46]. To further understand how ERα+ MPA neuronal activity affects this equilibrium, we monitored the temperature of the body core, BAT, and tail skin. Thermal probes were implanted intraperitoneally to measure core temperature, and thermal imaging of the interscapular region was used to monitor BAT thermogenesis. A temperature logger attached near the ventral vein of the tail, together with thermal imaging of the tail, were used to measure heat dissipation (Supplementary Fig. 1h). Infrared thermal images indicated a profound increase in tail skin temperature 20 min after CNO injection, whereas saline treatment was associated with a transient decrease in tail skin temperature, perhaps due to handling stress[47] (Fig. 1f, g). Consistent with regulated hypothermia, the rapid drop in core temperature (Fig. 1d) was not accompanied by an increase in heat generation, as indicated by both infrared imaging at multiple time points and gene expression in BAT measured 90 min following saline or CNO treatment (Fig. 1f, g and Supplementary Fig. 1i). Using thermo-loggers attached to the tail (Supplementary Fig. 1h), we observed similar changes in tail skin temperature and used these to estimate heat dissipation by calculating the heat loss index (HLI): $HLI = (T_{skin} - T_{ambient})/(T_{core} - T_{ambient})$[48]. HLI largely corrects for the effect of overall body cooling on tail skin temperature by comparing the temperature of the skin and core to the ambient temperature. Analysis of HLI revealed that CNO injection elicited a profound increase in heat loss that was coincident with the initial decrease in core temperature (Supplementary Fig. 1j, k). Consistent with previous studies showing that the POA mediates responses to warmth, including cutaneous vasomotor control[10,49], these results indicate that activating ERα+ MPA neurons initiates a rapid heat loss and suppresses heat production to induce hypothermia.

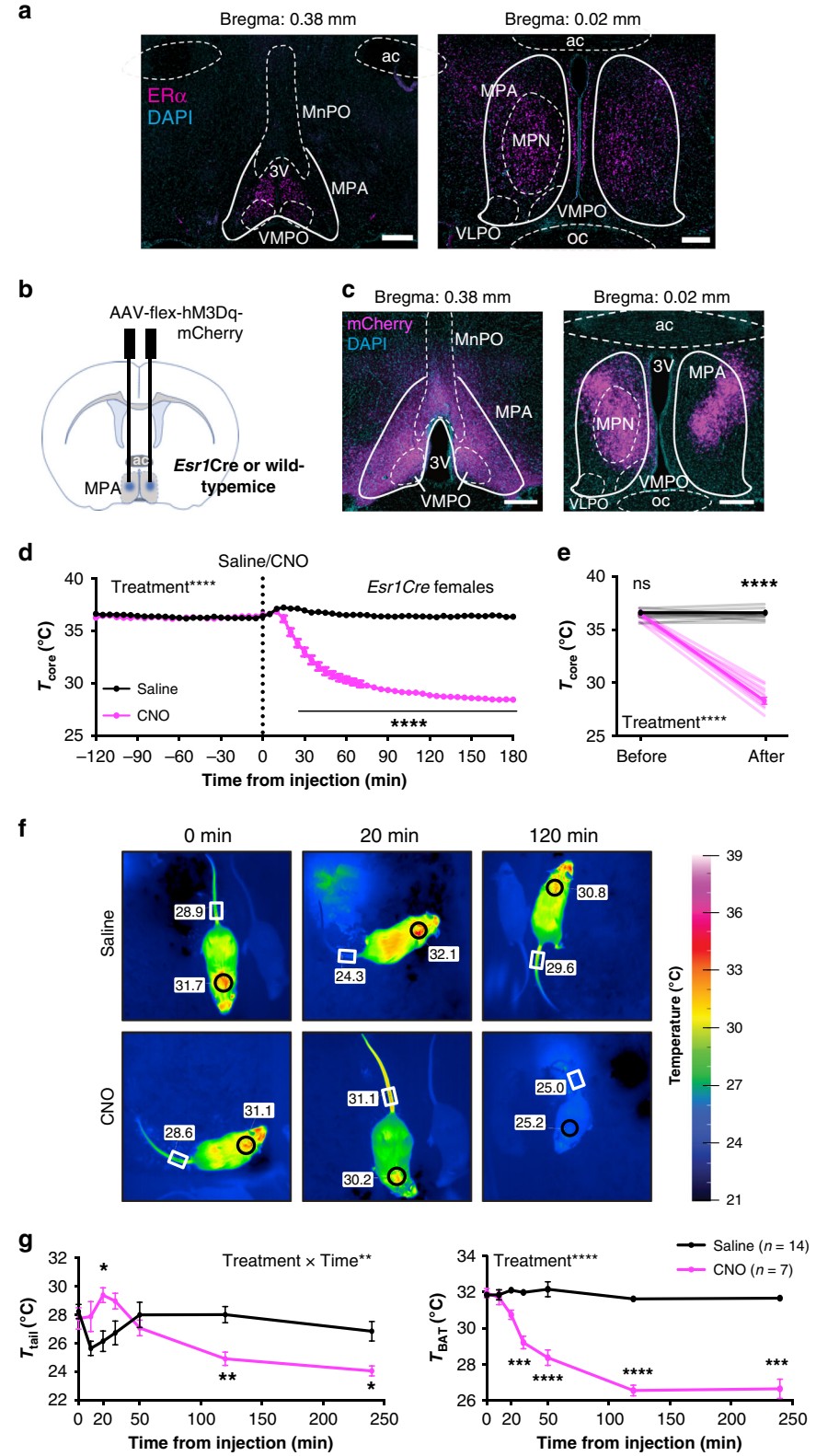

**Chemogenetic activation of ERα⁺ MPA neurons depresses temperature and metabolism**. Analysis of the changes in body temperature revealed that a single CNO injection could induce a state of prolonged hypothermia. Of the 42 *Esr1Cre* females expressing hM3Dq and treated with CNO, 31 mice (74%) exhibited a core body temperature below 31 °C and 12 mice (29%

of all subjects) maintained this level of hypothermia for over 20 h (Supplementary Fig. 2a, b). Indeed, core temperature returned to baseline (36.5 °C) up to 4 days after a single CNO injection (Fig. 2a). The reduction in core temperature also was accompanied by diminished physical activity (Fig. 2b). The expression of torpor has been previously associated with body weight[50–52].

**Fig. 1 ERα⁺ neurons in the MPA drive a profound decrease in core body temperature. a** ERα immunoreactivity (magenta) in the MPA of adult female mice. Image representative of $n = 4$ mice. Scale bar: 200 μm. 3v third ventricle, ac anterior commissure, MnPO median preoptic nucleus, MPA medial preoptic area, MPN medial preoptic nucleus, oc optic chiasm, VLPO ventrolateral preoptic nucleus, VMPO ventromedial preoptic nucleus. **b** Schematic of the AAV encoding a Gq-coupled DREADD (AAV-flex-hM3Dq-mCherry) delivered to the MPA of Esr1Cre mice. **c** mCherry reporter expression in the MPA of Esr1Cre females. Image representative of $n = 13$ mice. 3 v third ventricle, ac anterior commissure, oc optic chiasm. Scale bar: 200 μm. **d** Core body temperature measured every 5 min before and after injection (dotted line at $x = 0$) of saline (black) or CNO (pink) in Esr1Cre females ($n = 13$). **e** Per-animal averages of core temperature before (−120 min to 0 min) and after (120 min to 180 min) saline or CNO injection ($n = 13$). **f** Representative infrared thermal images showing temperature above BAT (circles) and tail (rectangles) after saline or CNO injection. **g** Quantification of thermography images in regions above the BAT and tail before and after saline ($n = 14$) or CNO ($n = 7$) injection in Esr1Cre female mice. *$p < 0.05$; **$p < 0.01$; ***$p < 0.001$; ****$p < 0.0001$ for Sidak's multiple comparison tests comparing saline and CNO following a significant effect of treatment or treatment time in a two-way RM ANOVA or Mixed-effects model. All error bars show SEM.

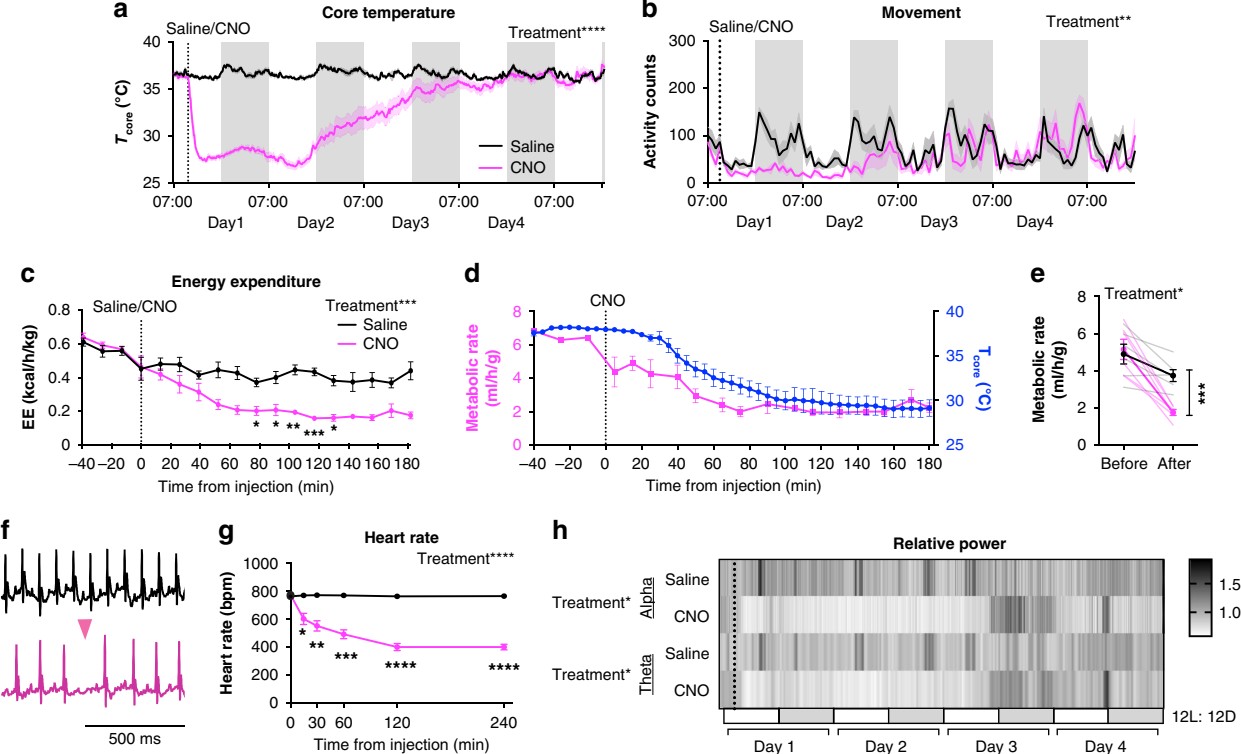

**Fig. 2 Chemogenetic activation of ERα⁺ MPA neurons induces a torpor-like state.** AAV-flex-hM3Dq-mCherry was delivered to the MPA of female Esr1Cre mice. **a**, **b** Four-day consecutive recordings of core temperature and movement after saline (black, $n = 8$) or CNO (pink, $n = 6$) injection. Lines show group means, shaded area shows SEM. Results of pairwise comparisons for **a**, **b** are listed in Supplementary Files. **c** Energy expenditure (EE or Heat) calculated from indirect calorimetry data and normalized to lean body mass following saline or CNO injection ($n = 6$). **d** Comparison of CNO-induced changes in metabolic rate (VO2 ml/h/g, pink) and Tcore (blue) in the same mice ($n = 4$ Esr1Cre females). **e** Metabolic rate (VO2 ml/h/g) before (−40 min to 0 min) and after (120 min to 180 min) saline or CNO injection calculated from indirect calorimetry data and normalized to lean body mass ($n = 6$ Esr1Cre females). **f** ECG traces in mice 4 h after saline or CNO injection. Traces representative of $n = 8$ mice. Pink arrowhead indicates a skipped heartbeat. **g** Heart rate ($n = 8$) before (0 min) to 4 h after saline or CNO injection. **h** Heatmap of relative alpha and theta power from four-day consecutive EEG recording after saline or CNO injection (dash line, $n = 4$). White and gray bars below represent light and dark periods, respectively. Group means ± SEM shown. *$p < 0.05$; **$p < 0.01$; ***$p < 0.001$; ****$p < 0.0001$ for Sidak's multiple comparison tests comparing saline and CNO following a significant effect of treatment **a** ($p < 0.0001$), **b** ($p = 0.0020$), **c** ($p = 0.0006$), **e** ($p < 0.05$), **g** ($p < 0.0001$), and **h** (Alpha $p = 0.04$ and theta $p = 0.02$) in a two-way RM ANOVA.

We also find that body weight was significantly correlated with the minimum core temperature reached and with duration of the hypothermia following CNO administration (Supplementary Fig. 2b). Even in warmer conditions (ambient temperature of 30 °C) that might diminish thermogenesis[45,53,54], activating ERα⁺ MPA neurons resulted in a rapid decrease in core temperature (Supplementary Fig. 2c), suggesting that reduced metabolic rate may also contribute to the reduction in body temperature. Indeed, indirect calorimetry revealed a reduction in metabolic rate and energy expenditure, as measured by oxygen

consumption and calculated heat production, respectively. Both measures were reduced by >50% in Esr1Cre mice following CNO injection, compared to the effects of saline injection in the same mice and to baseline levels before injection (Fig. 2c–e and Supplementary Fig. 2d). We did not detect an effect of CNO on respiratory exchange ratio (RER), a measure of fuel preference, within 3 h of injection. However, RER seems to decline in both groups, consistent with measurement during a daytime fast (Supplementary Fig. 2e). As further indication of an overall reduction in metabolic rate, electrocardiogram recordings

revealed a significant reduction in heart rate as early as 15 min after CNO injection, dropping from 776 ± 3.2 (mean ± SEM) bpm at baseline to 401 ± 18.8 bpm 4 h after CNO injection. Control saline injection did not affect heart rate (Fig. 2f, g and Supplementary Fig. 2g). The CNO-induced reduction in heart rate was frequently accompanied by pronounced sinus arrhythmia (Fig. 2f and Supplementary Fig. 2g), a known effect of enhanced cardiac vagal tone[55] that was never observed after saline treatment. These results suggest that the cardiac autonomic nervous system may be involved in this coordinated response. Finally, electro-encephalography (EEG) was used to measure overall brain activity during hypothermia and hypometabolism. CNO induced reductions of relative alpha and theta power in the EEG signal (Fig. 2h). Comparisons of relative delta power following saline or CNO injection revealed significant differences from non-rapid eye movement (NREM) or wake states (Extended Fig. 2h). Reductions such as these are common in experimental models of hibernation and torpor (see ref. [56] for review). The induction of this phenotype was repeatable, and mice seemed to recover without adverse impacts on their health (Supplementary Fig. 2i, j). Taken together, these findings indicate that activation of ERα+ MPA neurons induces a torpor-like hypothermic and hypometabolic state in mice.

**ERα+ neuronal activity increases during fasting-induced torpor.** The state triggered by activating ERα+ neurons suggests the hypothesis that ERα+ neuron activity is modulated during natural hypothermic and hypometabolic states. Prolonged food deprivation has been shown to induce torpor in mice[57]. Here, a 48 h fast resulted in consistent torpor bouts (Supplementary Fig. 3a). To record neural activity from ERα+ MPA neurons in live animals during fasting-induced torpor, we targeted the high-sensitivity sixth-generation, slow-kinetic calcium reporter GCaMP6s[58] to ERα+ neurons by delivering Cre-dependent AAV9-FLEX-Syn-GCaMP6s to the MPA of Esr1Cre mice (Fig. 3a). Following a 3-week recovery, mice were fasted for 48 h and fluorescence from ERα+ MPA neurons was measured during the last 8 h of fasting. Calcium transients were measured over eight 10-minute periods, 4 h before and 4 h after lights on, when torpor bouts are more likely to occur (Fig. 3b). Fasting-induced bouts of reduced core body temperature were consistent with fasting-induced torpor (Fig. 3c and Supplementary Fig. 3a). We detected larger and more variable calcium transients during bouts of hypothermia (core temperature < 33 °C) compared to times of normothermia (core temperature > 36 °C) following a fast (Fig. 3d, e). The decrease in baseline activity during hypothermia is in line with recently reported neuronal activity changes in torpor neurons[39]. Despite the reduced baseline, which may suggest a repression of tonic firing, the large peaks during hypothermia are consistent with an increase in synchronized burst firing. These results suggest a relationship between the natural activity of ERα+ MPA neurons and bouts of fasting-induced torpor.

Recent studies have identified neurons and molecular markers in the POA that are responsive to changes in temperature and involved in thermoregulation[38,59–62]. To exclude the alternative hypothesis that ERα+ MPA neurons are responsive to warmth, primarily involved in driving cooling responses, and that the drop in metabolism is secondary to hypothermia, we measured temperature responsiveness using calcium imaging and electro-physiology. We did not detect changes in calcium transients in freely behaving mice exposed to warmth (40 °C) or cold (15 °C), suggesting that changes in ambient temperature do not alter the neural activity of the ERα+ MPA neuron population (Supplementary Fig. 3b–e). Although we were unable to detect a response to temperature at the population level, it is possible that

temperature alters neural activity at the level of individual ERα+ MPA neurons. Thus, we recorded neural activity in brain slices from ERα-ZsGreen mice, in which ERα+ cells are identified by green fluorescence[63] (Fig. 3f). Whole-cell current clamp record-ings revealed heterogeneity in the temperature responsiveness of ERα+ neurons (Fig. 3g, h and Supplementary Fig. 3f, g). Using a 41.4% change in firing rate over a 5 °C temperature increase (Q10 > 2)[64,65] as a cutoff for temperature responsiveness, approximately half (56.5%) of the 108 ZsGreen+ MPA neurons were considered temperature responsive. The ERα+ MPA neurons tested include ZsGreen+ neurons in the medial preoptic nucleus (MPN, 56.2% of 73 neurons) and rostral aspects of the MPA (57.1% of 35 neurons) (Fig. 3h and Supplementary Fig. 3g). In contrast, ZsGreen+ neurons in adjacent nuclei such as the VLPO and VMPO contained only 13.9% and 28.6% temperature-responsive ZsGreen+ neurons, respectively, suggesting heterogeneity of ERα+ neurons with respect to temperature responsiveness (Supplementary Fig. 3g).

Consistent with temperature responsiveness within some ERα+ MPA neurons, RNA sequencing analysis of cellular contents following electrophysiological recordings revealed higher expression of warmth-induced genes[66] in temperature-responsive compared to non-responsive neurons (Fig. 3i). Global expression analysis of ERα+ neurons in the MPN after electrophysiology recordings revealed a strong separation of transcriptional signatures between temperature-responsive and non-responsive neurons (Supplemen-tary Fig. 3h), indicating that these two ERα+ populations are transcriptionally distinguishable, possibly due to either their function in temperature responsiveness or changes induced by exposure to 30 °C. Whereas ERα+ cells in the MPA include glutamatergic and GABAergic populations[14], temperature-responsive ERα+ neurons show enriched expression of GABAergic transcripts and reduced expression of glutamatergic transcripts when compared to non-responsive ERα+ neurons (Supplementary Fig. 3i). Additionally, temperature-responsive ERα neurons show enriched expression of the monoamine transporter Slc18a2, which is differentially expressed in a subpopulation of ERα neurons identified by high-resolution spatial RNA profiling[14] (Supplemen-tary Fig. 3i). Thus, temperature responsiveness was not detected in vivo and only detected in a GABAergic Slc18a2+ subpopulation of ERα+ neurons ex vivo. However, both temperature-responsive and non-responsive ERα+ neuron populations in the MPN express markers of neurons that are activated during torpor in mice[39] (Supplementary Fig. 3i).

**ERα+ MPA neurons are required for thermoregulatory homeostasis and the full expression of torpor.** To determine if ERα+ MPA neurons are required for temperature homeostasis or fasting-induced torpor, we selectively ablated ERα+ neurons in adult mice using AAV that expresses a genetically modified cas-pase 3 (Fig. 4a). This approach has been shown to effectively delete cells in vivo by triggering cell-autonomous apoptosis[67]. AAV2-FLEX-Caspase3 was delivered to the MPA of Esr1Cre mice. Controls included wild-type mice receiving AAV encoding the Cre-dependent caspase 3 or Esr1Cre mice receiving AAV encoding a Cre-dependent GFP. Four weeks after AAV delivery, Esr1Cre mice showed a >40% reduction in ERα immunoreactivity in the MPA compared to controls (Fig. 4b, c). Ablating ERα+ cells in the MPA led to a significant increase in core temperature in female but not in male mice (Fig. 4d, e). In contrast, neuron ablation did not affect physical activity in either female or male mice, suggesting that the temperature increase was selective and not a consequence of changes in movement (Supplementary Fig. 4a–d). The effect on core temperature indicates that ERα+

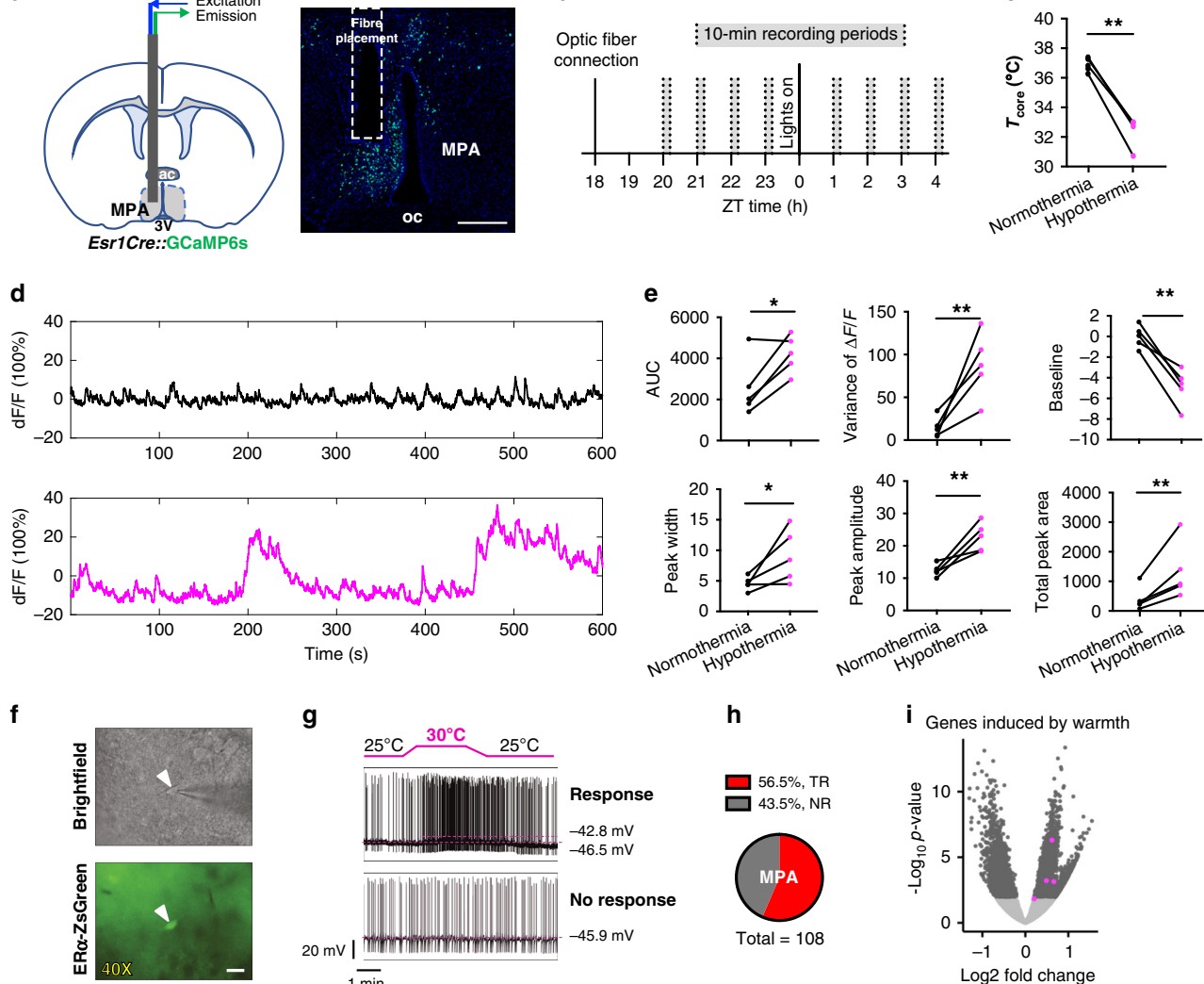

**Fig. 3 ERα⁺ neuronal activity during fasting-induced torpor in vivo and warmth exposure ex vivo. a** Schematic and coronal sections showing the optical fiber placement to record ERα⁺ neuronal activity in the MPA in vivo during fasting-induced torpor bouts. Image representative of $n = 5$ mice. oc optic chiasm. Scale bar 200 um. **b** Diagram showing the experimental design for photometry recordings during fasting. Mice were food deprived for 38 h at the time of optic fiber connection. **c** Core body temperature in mice showing normothermia and hypothermia (<33 °C) during the recording period following a fast. **d** Representative fluorescence traces of an individual mouse showing normothermia (black) and hypothermia (pink) during a 10 min recording period. **e** Mean area under the curve (AUC), variance, baseline, base width of peaks, peak amplitude, and total peak area of fluorescence detected from ERα⁺ MPA neurons in fasted mice showing normothermia (black) and hypothermia (pink) during a 10-min recording period ($n = 5$ mice, 4 females and 1 male). **f** Brightfield and fluorescence imaging of ERα-ZsGreen reporter expression in the MPA. Images representative of 179 cells from 4 mice. Scale bar 20 μm, white arrowhead denotes cell of interest. **g** Representative whole-cell current clamp recordings of ERα neurons in the MPA exposed to 25 °C and 30 °C. **h** Pie chart showing percentages of neurons showing a temperature response (TR, > 41.4% increase in firing rate) or no response (NR) in the MPA (MPN and rostral MPA). **i** Volcano plots comparing gene expression in TR ($n = 4$ samples, 2 neurons per sample) versus NR ($n = 3$ samples, 2 neurons per sample) ERα neurons in the MPN and rostral MPA regions combined. All genes shown as dots, with color denoting not significant in two-tailed differential expression testing (adjusted $p > 0.05$ using the Benjamini–Hochberg procedure, gray), significant (adjusted $p < 0.05$, dark gray), and genes induced by warmth[66].

MPA neurons are critical for maintaining normal thermal homeostasis in female but not in male mice.

We next investigated if ERα⁺ MPA neurons are required for fasting-induced torpor. Food deprivation induced bouts of hypothermia ($T_{core} < 31$ °C) in mice with intact ERα⁺ MPA neurons (Fig. 4f). Ablating ERα⁺ neurons in mice of similar body weights (Extended Fig. 4e) greatly diminished the torpor response, affecting the changes in average core temperature, core temperature variability, total time with a core temperature below 31 °C, duration of the longest bout, and lowest core temperature reached in fasted mice (overall effect of treatment, Fig. 4g and

Supplementary Fig. 4e). We did not detect a significant effect of neuron ablation on the number of bouts initiated, suggesting that ERα⁺ MPA neurons are required for maintaining rather than initiating torpor bouts. Comparisons by sex suggest that ERα⁺ MPA neurons may regulate torpor differently in females and males. We detected a significant effect of sex on core temperature variability, duration of the longest bout, and the lowest core temperature reached following a fast (Fig. 4g and Supplementary Fig. 4e). In pairwise comparisons within a sex, core temperature variability was significantly lower in females with neuron ablation but not in males (Fig. 4g). Additionally, the effect of neuron

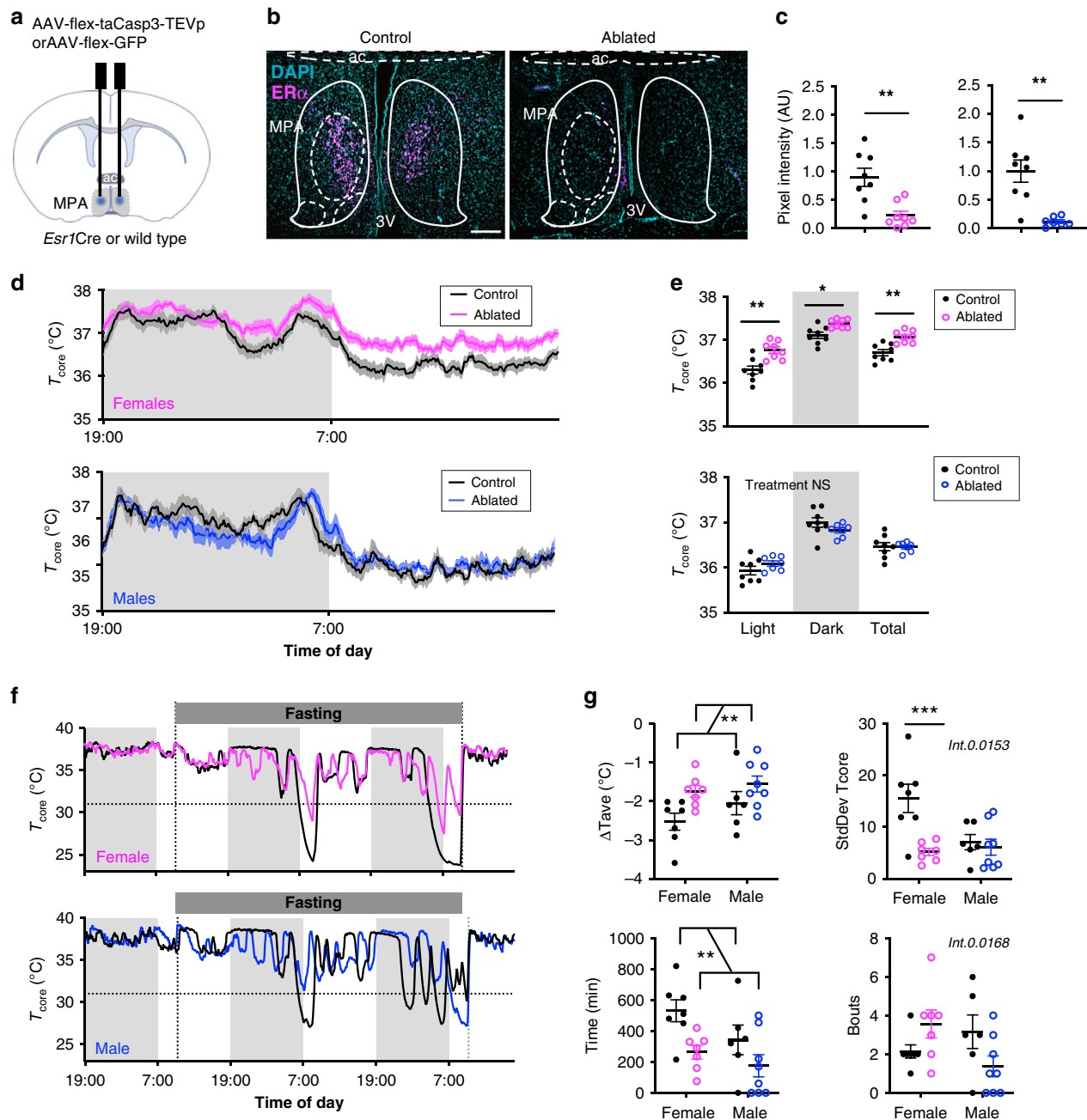

**Fig. 4 ERα⁺ cells in the MPA are required for thermoregulation and fasting-induced torpor. a** Schematic showing stereotaxic delivery of AAV encoding a Cre-dependent caspase to the MPA of *Esr1Cre* mice to ablate *Esr1*-expressing cells. **b** ERα immunoreactivity in the MPA of female mice following AAV-mediated ablation of *Esr1*-expressing cells (right, representative of $n = 8$ mice) or littermate controls (left, representative of $n = 8$ mice). 3v third ventricle, ac anterior commissure. Scale bar: 200 μm. **c** ERα immunoreactivity in the MPA of $n = 8$ control females (left plot, black circles), $n = 8$ ablated females (pink circles), $n = 8$ control males (right plot, black circles), and $n = 7$ ablated males. **d** Core body temperature ($T_{core}$) over 24 h, measured every 5 min for 3 days. Group averages shown for control (black, $n = 8$) and ablated (pink, $n = 8$ female; blue, $n = 7$ male) mice. Shading along the curve denotes the SEM. **e** Average $T_{core}$ from mice shown in panel **d** highlighting per animal averages in light (7:00 to 19:00), dark (19:00 to 7:00), and total 24 h periods. **f** Representative $T_{core}$ measured every 5 min over 3 days during fasting-induced torpor in control (black) and ERα neuronal ablated (pink) female mice. Dashed line denotes $T_{core} = 31°C$, the cutoff for hypothermia used here **g** Change in $T_{core}$ ($\Delta T_{core} = $ Lowest $T_{core}$ - Avg $T_{core}$ before the fast), variace of the $T_{core}$, total time spent with $T_{core} \leq 31°C$, and number of bouts with $T_{core} \leq 31°C$ during the fasting period ($n = 7$ control females, 7 ablated females, 6 control males, 8 ablated males). Statistical significance denoted by NS, not significant; *$p < 0.05$; **$p < 0.01$; ***$p < 0.001$ for two tailed t-tests **c**, Sidak's multiple comparison tests **e**, **g** following a significant effect of treatment, time or sex or interactions in a two-way (RM) ANOVA. All error bars show SEM.

ablation on the number of bouts initiated varied by sex (interaction effect, Fig. 4g). While it appeared that neuronal ablation lead to more bouts initiated in females and fewer bouts initiated in males, pairwise comparisons did not reach statistical significance (Fig. 4g). Together, these findings suggest that ERα⁺

MPA neurons are necessary for the full expression of fasting-induced torpor. Specifically, ERα⁺ MPA neurons are critical for maintaining the duration of torpor in all sexes but may also have sex-dependent roles in the variability of body temperature during torpor and the initiation of torpor bouts.

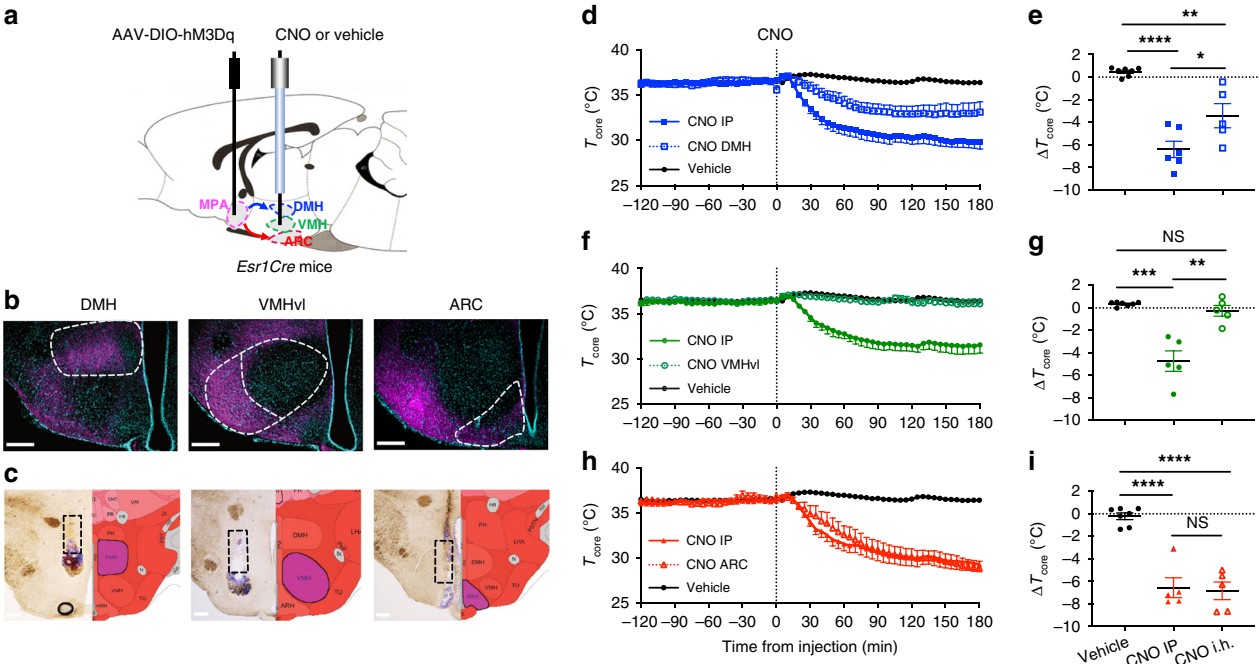

**Fig. 5 Activation of estrogen-sensitive hypothalamic circuits drive decreases in core body temperature. a** Schematic strategy for testing the thermoregulatory hypothalamic circuits from the MPA by intrahypothalamic (i.h.) delivery of CNO or vehicle (artificial cerebrospinal fluid, aCSF) to the projected nuclei. **b** mCherry fibers indicating projections from MPA ERα+ neurons to DMH (representative of $n = 6$ mice), VMHvl (representative of $n = 5$ mice), and ARC (representative of $n = 5$ mice). ARC arcuate nucleus, DMH dorsal medial hypothalamus, MPA medial preoptic area, VMHvl ventral lateral subregion of ventromedial hypothalamus. Scare bars: 250 µm. **c** Brightfield coronal sections (left) and diagram coronal sections (right) showing the canular track and targeted nuclei for local CNO delivery. **d**, **f**, **h** Core temperature ($T_{core}$) measured every 5 min before and after injection (dotted line at $x = 0$) of vehicle (saline and aCSF data combined, $n = 7$), CNO IP (0.3 mg/kg) or CNO i.h. (8.4 ng/mouse) in the DMH (blue, $n = 6$ mice shown in **d** and **e**), VMH (green, $n = 5$ mice shown in **f** and **g**), and ARC (red, $n = 5$ mice shown in **h** and **i**). **e**, **g**, **i** Change in core temperature ($\Delta T_{core}$) after vehicle or CNO administration (120 min to 180 min after injection) compared to baseline (−120 min to 0 min before injection). Statistical significance denoted by NS, not significant; *$p < 0.05$; **$p < 0.01$; $p < 0.01$; ***$p < 0.001$, and ****$p < 0.0001$ for posthoc two-tailed Tukey's multiple comparisons test following a significant effect of treatment ($p < 0.0001$) in a one-way ANOVA.

**ERα+ MPA neurons drive hypothermia through projections to the ARC and DMH.** To visualize the downstream projections of ERα+ neurons from the MPA, we performed anterograde tracing using a Cre-dependent AAV expressing GFP (Supplementary Fig. 5a). We detected dense GFP+ fibers in several hypothalamic nuclei involved in thermoregulation, including the dorsomedial nucleus of the hypothalamus (DMH), ventromedial nucleus of the hypothalamus (VMH), and the arcuate nucleus of the hypothalamus (ARC) (Supplementary Fig. 5b, c). The DMH receives direct preoptic inputs[38,62] and modulates the sympathetic outputs to BAT in response to cold and warm ambient temperature[44]. The VMH has been implicated in estrogenic regulation of BAT thermogenesis[31] and many estrogenic effects on energy metabolism[33,37]. The ARC contains neurons implicated in the hormonal modulation of vasodilatation during hot flashes through projections to the POA[68]. To test the contribution of projections from ERα+ MPA neurons to these regions, we repeated the chemogenetic hM3Dq activation experiments but delivered the CNO locally to the DMH, VMH, or the ARC (Fig. 5a). This method exploits trafficking of hM3Dq to axon terminals to alter presynaptic activity and is effective both ex vivo and in vivo[69,70]. In line with the results of the anterograde tracing (Supplementary Fig. 5b, c) and previous studies[69], we observed numerous mCherry positive fibers in the downstream target sites within the hypothalamus (Fig. 5b). We first confirmed that systemic administration of CNO by IP injection reduced core body temperature as in Fig. 2. When CNO was delivered to the DMH, core temperature was significantly reduced compared to delivery of vehicle but did not recapitulate the effects of systemic CNO

administration (Fig. 5c–e). CNO delivery to the VMH had no effect on body temperature and was similar to vehicle injection (Fig. 5f, g). In contrast, local delivery of CNO to the ARC led to a reduction in body temperature that was indistinguishable from the effect of systemic CNO injection in the same mice (Fig. 5h, i). The effects of delivering CNO to distinct projection targets of ERα+ MPA neurons suggest that projections to the ARC, and to some extent DMH, could mediate the effects of ERα+ MPA neurons.

## Discussion

Torpor has been investigated in numerous species, yet the neuronal mechanisms allowing endotherms to actively depress body temperature and metabolism in response to nutrient scarcity are only fragmentally understood[1,3]. Here we exploit the viral and genetic tools available in mice to show that selective activation of ERα+ MPA neurons induces a coordinated depression of metabolic rate, body temperature, physical activity, heart rate, respiratory rate, and brain activity. In the majority of cases, the induced torpid state lasted from 20 h to multiple days, far outlasting the pharmacokinetic duration of CNO (1–3 h[71], although the duration could be longer if torpor also reduces CNO metabolism). The state induced by activating ERα+ MPA neurons resembles entry into torpor in several ways. As observed in several rodent species during entry into daily torpor[72–74], we observe that the drop in metabolic rate declines faster than core temperature. We also observe a rapid drop in heart rate and characteristic skipped heart beats, as observed in multiple small mammalian species[75,76], along with reduced theta power brain

activity, as observed in ground squirrels entering torpor[77]. Additionally, we observe variation in the minimum core temperature, variation in the duration of hypothermia, and an association between the minimum body temperature and body weight, which have been reported in spontaneous torpor[50]. Indeed, individual variation in torpor expression has been well documented[50,57,74] and associated with birth weight[51] or body weight at the time of fast[52] in mice. Overall, the state induced by activating ERα+ MPA neurons recapitulates several key facets of torpor, including hypometabolism, hypothermia, reduced physical activity, and reduced brain activity in fed mice.

The relationship between body weight and the hypothermic response to activating ERα+ MPA neurons suggests that nutritional experience or metabolic signals may alter the neural circuits that regulate torpor. In line with a proposed effect of nutritional signals, activating POA neurons that express the receptor for leptin, a signal of fat stores, also induces hypothermia[61] and modulating leptin receptor signaling in the POA alters energy expenditure[78]. In addition, several molecules have been employed to induce torpor-like hypothermia, including nucleotides (adenosine, cyclohexyladenosine, AMP, and ADP), anesthetics, and 3-Iodothyronamine (T1AM)[76,79–82]. Central activation of the A1 adenosine receptor also produces a torpor-like state in rats[82] but ablation of the adenosine A1 receptor does not affect entrance into torpor[83]. Although a detailed mechanism is unknown, the duration of DREADD-induced torpor suggests that ERα neurons may act as a "switch" to turn on torpor by triggering a neural circuit that mediates an inverted thermoregulatory state[84].

Beyond the sufficiency of ERα+ MPA neurons to induce torpor, we examined the natural activity and the necessity of ERα+ MPA neurons in an endogenously regulated torpid state induced by fasting in mice[57]. Calcium imaging and neuron ablation studies suggest that ERα+ MPA neurons are activated during bouts of fasting-induced torpor and are important for the full expression of torpor in response to fasting. Our results suggest that ERα+ MPA neurons are involved in maintaining torpor in both female and male mice, and may also have a role in initiating torpor bouts that varies across the sexes. Ablating ERα+ MPA neurons appeared to increase the number of torpor bouts in females and decrease the number of torpor bouts in males. Although this effect was not significant in pairwise comparisons within a sex, there was a significant interaction between sex and neuron ablation. The trend in females could be explained by compensation, such that female mice with shorter torpor bouts might initiate more bouts to conserve energy. The trend in males is surprising, because they did not appear to initiate more bouts to compensate for shorter bout durations. This sex difference is consistent with evidence that torpor is regulated differently in males and females, and that females are generally more "thrifty" with their energy reserves than males[85]. Indeed, in several rodent species, females initiate hibernation earlier, have more or longer torpor bouts, spend more time in torpor, and terminate hibernation later than males[86–90].

Ablating ERα+ MPA neurons also led to a female-specific effect on basal body temperature. This result is consistent with previous studies showing effects of estrogen-sensitive neurons on body temperature in female mice[91] and with evidence that temperature-responsive neurons in nearby regions of the POA regulate blood flow to the skin[7,92,93] and BAT activity[12,38,62]. Indeed, a GABAergic subset of ERα+ neurons in the MPN appear to be temperature responsive with respect to firing and gene expression changes. Although this responsiveness to temperature may not be strong enough to drive a response that is detected by calcium imaging from the whole population in vivo, it is consistent with previous findings of warm-responsive GABAergic

neurons in the POA[38,62] but different from warm-responsive glutamatergic POA neurons that decrease body temperature when activated in mice[60,61,91,94]. In total, about ~30% of neurons in the POA are temperature responsive[64,95]. Interestingly, approximately half of the warm-responsive neurons in the POA are responsive to estradiol, compared to only one-third of temperature-unresponsive neurons[96]. However, we do not have any evidence to implicate or exclude a role for ERα signaling in torpor. Future studies will determine if torpor neurons are modulated by changes in estrogen levels or estrogen receptor manipulations.

We also find that temperature-responsive ERα+ neurons differentially express Slc18a2, suggesting that an inhibitory cluster of ERα+ Slc18a2+ neurons in the MPN[14] may be responsive to warmth and perhaps drive changes in body temperature. However, it is unclear if this neuron population mediates the effects on basal body temperature or the torpor response. Indeed, activating ERα+ MPA neurons induced rapid changes in tail skin vasodilation and inhibition of BAT thermogenesis. Because mice are highly dependent on somatic activity for thermoregulation, it is possible that the reduction in physical activity, inhibition of BAT thermogenesis, stimulation of heat dissipation at the tail, and bradycardia may together contribute to the rapid decrease in core body temperature observed in torpor. These effects may be mediated by modulation of multiple neural targets of the MPA or hormonal mediators such as thyroid hormone[97]. However, evidence that the drop in metabolic rate precedes hypothermia suggests that the multifaceted changes induced by activating ERα+ MPA neurons are not secondary to thermal changes (Q10)[74]. Because ERα+ neurons in the MPN appear to include temperature-responsive and non-responsive neurons as well as GABAergic and glutamatergic neurons, it will be important to dissect the contributions of the different subpopulations (i.e., ERα+ Slc18a2+ and ERα+Slc18a2− neurons) to thermoregulation and torpor.

Pioneering studies have identified hypothalamic nuclei and circuits that are critically involved in regulation of body temperature and metabolism[9,12,98]. Our tracing studies reveal projections from ERα+ neurons to several of these hypothalamic nuclei, including the DMH, VMH, and ARC. Local activation of ERα+ neurons that project to the DMH led to a partial decrease in core body temperature, whereas local activation of ERα+ neurons that project to the ARC recapitulated the effect of systemic CNO delivery on core body temperature. These findings are consistent with evidence that projections from the POA to the DMH regulate thermogenesis, heart rate, and energy expenditure[38,62] and an increase in cFOS immunoreactivity within the DMH during fasting-induced torpor[98]. Similarly, the ARC is implicated in the regulation of heat dissipation[68,99], thermogenesis[100,101], and fasting-induced torpor[98,102]. It is possible that locally delivered CNO may diffuse to adjacent hypothalamic regions. For example, the partial effect of delivering CNO to the DMH could also be due to some diffusion to the ARC or elsewhere. However, CNO delivery to the VMH did not affect body temperature, excluding a role for the VMH in this circuit and suggesting that diffusion of CNO to adjacent regions may have been limited in these experiments. Additionally, CNO-induced activity may cause retrograde conduction and activate collateral projections. Although differential effects were observed when delivering CNO to the ARC, DMH, and VMH, we cannot exclude the possibility that ERα+ MPA neurons project to multiple regions and induce torpor via collateral projections. Considering the autonomic changes (cardiac function, BAT inhibition, and tail vasodilation) observed here, it is possible that ERα neurons also project to the raphe pallidus[40,92,103] for the autonomic coordination of torpor. Although more selective

circuit mapping and manipulations are required to fully understand the circuits that control torpor, these studies provide additional evidence of a potent role for ERα+ MPA neurons in the regulation of torpor.

In summary, these studies identify a neuron population that is critical for an adaptive response that minimizes energy costs and increases survival during food shortages in mice[74]. These findings are consistent with very recent studies showing that a hypothermic and hypometabolic state can be induced by activating POA neurons, specifically glutamatergic neurons that express pyroglutamylated RFamide peptide (*Qrfp*)[40] or neurons expressing adenylate cyclase activating polypeptide 1 (*Adcyap1*)[39]. Correspondingly, we find expression of *Adcyap1* and other gene markers of torpor neurons in ERα+ MPA neurons and general overlap in projection target regions. A major role for the ARC but not the VMH in the regulation of torpor is consistent with evidence that torpor neurons in the POA strongly innervate the ARC but not the VMH[39]. However, QRFP neurons appear to exert their effects primarily through projections to the DMH[40], whereas our preliminary data suggest that ERα+ MPA neurons may exert their effects through projections to the ARC and partially through the DMH. Nonetheless, it is possible that the intersection of these three gene markers may define the neurons required for torpor or that multiple populations in this region together coordinate the torpor response.

## Methods

**Animals**. Mice harboring the *Esr1Cre* knock in allele (*Tm1.1(Cre)And*)[36] were maintained on a C57BL/6J genetic background and bred in approved facilities at University of California Los Angeles (UCLA). For all experiments using *Esr1Cre* mice, controls included Cre-negative (wild-type) littermates that received the same AAV as the experimental group and/or *Esr1Cre* mice receiving a control AAV, as described for each experiment. The ERα-ZsGreen transgenic mice[63] used for electrophysiology were maintained on a C57BL/6J genetic background and bred in approved facilities at Baylor College of Medicine. All mice were maintained under a 12:12 h L/D schedule at room temperature (~22 °C), maintained at 30–70% relative humidity, and provided with food and water ad libitum unless otherwise indicated. Mice were at 8–10 weeks at the start of all the experiments.

**Mouse procedures**. All studies were carried out in accordance with the recommendations in the Guide for the Care and Use of Laboratory Animals of the National Institutes of Health. UCLA is AAALAC accredited and the UCLA Institutional Animal Care and Use Committee (IACUC) approved all animal procedures. Mice were anaesthetized with isoflurane and received combinatorial analgesics (buprenorphine and carprofen) pre- and post any surgeries.

*Stereotactic surgery*. The pAAV-flex-taCasp3-TEVp was a gift from Nirao Shah & Jim Wells (Addgene plasmid # 45580). The pAAV-Syn-FLEX-Mac-GFP was a gift from Edward Boyden (Addgene plasmid # 58852). The pAAV-hSyn-DIO-hM3D (Gq)-mCherry and pAAV-hSyn-DIO-mCherry were gifts from Bryan Roth (Addgene plasmid # 44361 and # 65417)[104]. The pAAV9-Syn-FLEX-GCaMP6s-WPRE-SV40 was a gift from Douglas Kim & GENIE Project (Addgene viral prep # 100843-AAV9; http://n2t.net/addgene:100843; RRID:Addgene_100843). AAVs (300 nl for GFP antegrade tracing, 150 nl for Caspase 3, Gq-coupled DREADDs and GCaMP6s) were injected bilaterally (or unilaterally for GCaMP6s) under stereotaxis to the MPA (coordinates: AP 0.2, ML ± 0.35, DV −5.3 from the surface of skull) of adult *Esr1Cre* or wild-type mice. Unilateral photometry fiber was implanted using the same coordinates for virus injections. For cannulation, bilateral stainless-steel guide cannulas 26 G (Plastics One) were implanted bilaterally targeted 2 mm above the DMH (AP −1.80, ML ± 0.45, DV −3.6), VMH (AP −1.5, ML ± 0.65, DV −3.9) or 3 mm above the ARC (AP −1.9, ML ± 0.25, DV −3.3). The guide cannulas or photometry fiber were mounted on top of the head using dental cement anchored with 2 screws fixed on skull. For the electroencephalogram (EEG) experiment, two anterior electrodes (frontal and ground, AP + 1.5, ML 1.5) and two posterior electrodes (parietal and common reference, AP −2.5, ML 1.5) were connected with a head mount (integrated 2 × 3 pin grid array) and secured to the skull with dental acrylic.

*Temperature recording*. A G2 eMitter (Starr Life Sciences) was implanted in the abdominal cavity and attached to the inside of the body wall. Mice were singly housed in cages placed on top of ER4000 Energizer/Receivers. Nesting material was held constant to normalize behavioral temperature regulation. Gross movement and core body temperature were measured every 5 min using VitalView software (version 5, Starr Life Sciences). Tail skin temperature was monitored every 5 min

using a Nano-T temperature logger and analyzed with Mercury Software (version 5.7, Star-Oddi). The logger was attached to the ventral surface and 1 cm from the base of the tail in a 3D-printed polylactic acid collar modified from Krajewski-Hall et al[27].

*Chemogenetics*. In DREADDs experiments, mice received i.p. injections of CNO (0.3 mg/kg, Sigma-Aldrich) or vehicle (saline, 0.15% DMSO) 3 h after the onset of the light phase. Saline and CNO were administered in the same mice in a randomized balanced design. Core and tail skin temperature were monitored continuously throughout the experiment. As a control, the DREADD ligand Compound 21 (1 mg/kg, Cayman Chemical Company) or vehicle (saline, 1% DMSO) was administered i.p. following the same experimental procedure as CNO injection. For intrahypothalamic injection, CNO was prepared at 2 mM in aCSF. Before injection, the mice were connected to a 33 G Stainless-Steel internal cannular (Plastic One) that was attached to 1 ul Hamilton Syringes through 40 cm non-compressive silicone tubing. The internal cannulae were cut 2 mm (for DMH and VMH) or 3 mm (for ARC) below the guide cannula. CNO or vehicle (aCSF 3.4% of DMSO) was injected at 50 nl/side in 1 min. The mice were able to move freely during injection.

Infrared thermal images were captured using an Industrial camera VarioCAM® HD head 800 (InfraTec infrared LLC) before (t0) CNO or vehicle injection, then 10, 20, 30, 50, 120, and 240 min after injection. The infrared images were analyzed using software IRBIS 3.1 (InfraTec infrared LLC). BAT skin temperature was the average temperature of a circular region above interscapular BAT and tail skin temperature was the average temperature of a 1 cm line along the tail starting at 1 cm from the base of the tail. Electrocardiogram (ECG) was recorded using ECGenie system (Mouse Specifics Inc.). Heart rate and other cardiological parameters were analyzed using EzCG Signal Analysis Software (version 7.0, Mouse Specifics Inc.). Indirect calorimetry was performed in Oxymax metabolic chambers and acquired by Oxymax software (version 3.3, Columbus Instruments). Body composition was determined using nuclear magnetic resonance (NMR) (Mouse Minispec, Brüker Corporation). EEG acquisition was performed by polysomnographic software (Sirenia Acquisition version 2.1, Pinnacle Technologies, Lawrence, KS). Signals were amplified (10x) and high-pass filtered (0.5 s$^{-1}$) via a preamplifier. EEG signals were then further amplified, low-pass filtered with a 30 s$^{-1}$ cutoff and collected continuously at a sampling rate of 200 s$^{-1}$. For relative banded power, data were normalized to the total power and averaged as 5 min bins. For sleep definition, EEG/EMG waveforms were classified in 10-sec epochs as: (1) wake (low-voltage, high-frequency EEG; high-amplitude EMG); (2) NREM sleep (high-voltage, mixed-frequency EEG; low- amplitude EMG); or rapid-eye movement (REM) sleep (low-voltage EEG with a predominance of theta activity [6–10 s$^{-1}$]; very low amplitude EMG) by a trained observer. EEG epochs determined to have artifact (interference caused by scratching, movement, eating, or drinking) were excluded from the analysis. Artifact comprised less than five percent of all recordings used for analysis. When mice were injected with CNO, their unscored band frequencies were compared to the band frequencies of wake and NREM of when the same mice were injected with saline. Fast Fourier Transform was completed on all recordings to determine the power analysis of each waveform.

*Fasting-induced torpor*. Mice were individually housed and core body temperature was monitored continuously as described above. Following baseline measurements (food and water ad libitum), mice were placed in new cages and food was removed from the cages at 10 am. After 48 h of fasting, the mice received their food back or were euthanized for perfusion. Mice are closely observed for symptoms of dehydration, sickness, and immobility during fasting. For torpor induction during ERα ablation experiment, torpor bouts were defined when the core body temperature was equal to or below 31 °C[105]. For photometry experiments, fasting-induced hypothermia was defined when core body temperature was below 33 °C.

*Fiber photometry*. Three weeks after AAV injection and temperature probe implantation, the mice were food deprived for 34 h and habituated 4 h before the start of recording. To record during periods of hypothermia and normothermia for each mouse, fluorescence was recorded for 10 min, hourly for the 4 h before and 4 h after lights on (Fig. 3b). Photometry was performed as described previously[106]. Briefly, we used a 405 nm LED and a 470 nm LED (Thorlabs, M405F1 and M470F1) for the Ca$^{2+}$-dependent and Ca$^{2+}$-independent isosbestic control measurements. The two LEDs were bandpass filtered (Thorlabs, FB410-10 and FB470-10) and then combined with a 425-nm longpass dichroic mirror (Thorlabs, DMLP425R) and coupled into the microscope using a 495-nm longpass dichroic mirror (Semrock, FF495-Di02-25 × 36). Mice were connected with a branched patchcord (400 μm, Doric Lenses, Quebec, Canada) using a zirconia sleeve to the optical system. The signal was captured at 20 s$^{-1}$ (alternating 405 nm LED and 470 nm LED). To correct for signal artifacts of nonbiological origin (i.e., photobleaching and movement artifacts), custom Matlab (V9.5 R2008a) scripts leveraged the reference signal (405 nm), unaffected by calcium saturation, to isolate and remove these effects from the calcium signal (470 nm).

To test the effect of ambient temperature on ERα neuronal activity, the temperature of the mouse cage was manipulated using an iron plate and either a heat pad or ice block. Real-time temperature in the cage was monitored by digital

thermometer with the extended sensor attached on the bottom of the cage. Changes in ambient temperature were recorded and depicted in Supplementary Fig. 3b. Photometry recordings were analyzed only after the ambient temperature reached 40 °C or 15 °C, not during the transitions.

Calcium signal was evaluated over 10 min recording traces for both normothermia (when $T_{core} >= 33$ °C) or hypothermia ($T_{core} < 33$ °C). A custom Matlab script was made to analyze the area under the curve (AUC), standard deviation of the $\Delta F/F$ (Variance of $\Delta F/F$), average base width of the peaks (peak width), amplitude of the peaks measured from baseline (peak amplitude), and total peak area. A 1 min sliding window was applied to calculate the local baseline (10th percentile value) and standard deviation of the $\Delta F/F$ values[39].

**Electrophysiology**. We used the ERα-ZsGreen reporter to identify ERα-expressing neurons in slice. This mouse line has been used to study electrophysiological properties of ERα[+] neurons in multiple brain regions and shows higher ZsGreen reporter expression in the MPA of females than in males[29,63]. To identify ERα cells more broadly and provide a fuller picture of their responsiveness to temperature, we used only female ERα-ZsGreen mice. Mice (10–14 weeks old) were deeply anesthetized with isoflurane and transcardially perfused with a modified ice-cold sucrose-based cutting solution (pH 7.3) containing 10 mM NaCl, 25 mM NaHCO3, 195 mM sucrose, 5 mM glucose, 2.5 mM KCl, 1.25 mM NaH$_2$PO$_4$, 2 mM Na-Pyruvate, 0.5 mM CaCl$_2$, and 7 mM MgCl$_2$, bubbled continuously with 95% O$_2$ and 5% CO$_2$. Mice were decapitated and the entire brain was removed and immediately submerged in the cutting solution. Slices (250 μm) were cut with a Microm HM 650 V vibratome (Thermo Scientific). The brain slices containing the MPA region were obtained for each animal. The slices were recovered for 1 h at 34 °C and then maintained at room temperature (25 °C) in artificial cerebrospinal fluid (aCSF, pH 7.3) containing 126 mM NaCl, 2.5 mM KCl, 2.4 mM CaCl$_2$, 1.2 mM NaH$_2$PO$_4$, 1.2 mM MgCl$_2$, 5.0 mM glucose, and 21.4 mM NaHCO$_3$, saturated with 95% O$_2$ and 5% CO$_2$ before recording. Slices were transferred to a recording chamber and allowed to equilibrate for at least 10 min before recording. ZsGreen-labeled neurons in the MPA were visualized using epifluorescence and IR-DIC imaging on an upright microscope (Eclipse FN-1, Nikon) equipped with a movable stage (MP-285, Sutter Instrument). Patch pipettes with resistances of 3–5 MΩ were filled with intracellular solution (pH 7.3) containing 128 mM K-Gluconate, 10 mM KCl, 10 mM HEPES, 0.1 mM EGTA, 2 mM MgCl2, 0.05 mM Na-GTP, and 0.05 mM Mg-ATP. Recordings were made using a MultiClamp 700B amplifier (Axon Instrument), sampled with Digidata 1440A, and analyzed offline with pClamp 10.3 software (Axon Instruments). Series resistance was monitored during the recording, and the values were generally <10 MΩ and were not compensated. The liquid junction potential was +12.5 mV, and was corrected after the experiment. Data were excluded if the series resistance increased dramatically during the experiment or without overshoot for action potentials. Currents were amplified, filtered at 1000 s$^{-1}$, and digitized at 10,000 s$^{-1}$. Current clamp was engaged to test neural firing frequency and resting membrane potential at room temperature as reported[107,108]. For the temperature treatment, bath solution was heated by the in-line solution heater and maintained at 30 °C by the heating chamber (Warner). Temperature changes from 25 °C to 30 °C are sufficient to activate warm sensing neurons[109,110]. The values for firing frequency were averaged within 2-min bin at 25 °C or 30 °C. A neuron was considered warm responsive when there was a more than 50% increase in firing rate over a 5 °C temperature increase (Q10 > 2)[64,65].

*Patch-seq*. After recording, the cellular component of each neuron was captured into the electrode pipet by delivering a gentle negative pressure, and then transferred into a PCR tube as in ref. [29]. Two neurons from each cell type were pooled as one sample. RNA extraction was performed using SMART-seq V4 ultra-low input kit (Takara) and cDNA Library was built using Nextera XT DNA library preparation kit (Illumina). Samples were pooled and sequenced in a single lane of Illumina HiSeq 3000 (185 million reads over 7 samples for an average of 26.43 million reads per sample). Demultiplexed reads were aligned to the mouse genome (version mm10) using RNA STAR (Galaxy version 2.6.0b-1). PCR duplicates were removed using RmDup (Galaxy version 2.0.1). Gene level counts were determined from BAM files using htseq-count (Galaxy version 0.9.1). Sample distances and differentially expressed genes were determined using DESeq2 (Galaxy version 2.11.40.6). Highlighted volcano plots were created using a custom R function (Volcano_Plot_GS) available at https://github.com/jevanveen/zhang.

**Immunohistochemistry**. Mice were perfused transcardially with ice-cold DEPC treated PBS (pH = 7.4) followed by 4% paraformaldehyde (PFA). Brains were embedded in OCT and frozen in −80 °C after one overnight post fixation in 4% PFA and another overnight dehydration in 30% sucrose. Coronal sections were cut using a cryostat (Vibratome) into 8 equal series at 25 μm for the GFP tracing experiment and 18 μm for the rest of the experiments.

*GFP, mCherry and GCaMP6s*. Sections were washed 1× for 5 min in PBS and incubated with DAPI (1:1000, Thermo Fisher Scientific) for 5 min. Slides were then coverslipped with Fluoromount-G (Thermo Fisher Scientific) after a 5-min PBS wash.

*ERα*. Sections were first incubated for 40 min at 95 °C in 25 mM Tris–HCl (pH 8.5), 1 mM EDTA, and 0.05% SDS (Tris-EDTA-SDS) buffer for antigen retrieval and then blocked for 1 h in 10% BSA and 2% normal goat serum (NGS). Next, the sections were incubated overnight at 4 °C with primary antibody (ERα, 1:250, sc-8002, Santa Cruz). Following 3× 10 min washing in PBS, sections were incubated with Alexa fluor 488 conjugated goat anti-mouse secondary antibody (1:500, Thermo Fisher Scientific) for 2 h at room temperature. After 2× 10 min washing in PBS, sections were incubated with DAPI, washed, and coverslipped with Fluoromount-G.

The images were taken by DM1000 LED fluorescent microscope (Leica) or LSM780 confocal microscope (Zeiss). Confocal images that contain tiles and z-stacks were stitched and merged by maximum intensity projections using Zen Black (version 2.3, Zeiss). Cyan/magenta/yellow pseudo-colors were applied to all fluorescent images for accessibility. Image processing was performed using the Leica Application Suite (version 4.10, Leica), Zen Black, and ImageJ (version 2.0, NIH). Quantification was performed using CellProfiler software (version 3.1.8, Broad Institute).

*RNA isolation and real-time PCR (qPCR)*. Interscapular BAT was dissected 90 min after CNO or saline injections. BAT tissue was then snap frozen in liquid nitrogen and stored at −80 °C until analysis. Total RNA from BAT was isolated using the Zymo RNA isolation kit (ZYMO Research) and RNA yield was determined using the NanoDrop D1000 (Thermo Fisher Scientific). cDNA synthesis was performed with equal RNA input using the Transcriptor First Strand cDNA synthesis kit (Roche Molecular Biochemicals). Quantitative PCR was performed using C1000 Touch Thermal Cycler (BioRad) and SYBR mix (Bioline, GmbH, Germany). The primers used are listed in Supplementary Table 1.

**Statistics**. Data are represented as mean ± standard error of the mean (SEM). Data with normal distribution and similar variance were analyzed for statistical significance using two-tailed, unpaired Student's *t*-tests. Paired data, such as within-subject comparisons, were analyzed by paired *t*-tests or ratio paired *t*-test. Comparisons for more than two groups were analyzed by one-way ANOVA followed by post-hoc Tukey's analysis. Time course data and sex difference data were analyzed by two-way ANOVA or repeated measures two-way ANOVA for paired data or mixed model followed by Sidak's multiple comparisons. Significance was defined at a level of $P < 0.05$. Plots were generated and statistical analyses were performed using GraphPad Prism version 8 or RStudio (using tidyverse version 1.3.0, nlme version 3.0, and R version 4.0).

## Data availability

The patch-seq data are deposited in the NCBI Gene Expression Omnibus (accession #GSE153350). All raw data generated during these studies (images, videos, phenotyping measures at finer time scales or during acclimations) are freely available from the corresponding author upon request. These will be provided as needed so that they can be delivered in the form, scale, and resolution that is requested. Source data are provided with this paper.

## Code availability

Custom R and MATLAB scripts are available at https://github.com/jevanveen/zhang.

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

## Acknowledgements

This research was supported by funds from the UCLA Division of Life Sciences and grant R01 AG066821 to S.M.C., a Pilot Award from the Iris Cantor-UCLA Women's Health Center/UCLA National Center of Excellence in Women's Health and NIH National Center for Advancing Translational Science (NCATS) UCLA CTSI Grant Number UL1TR001881 to Z.Z. and S.M.C., an American Heart Association Postdoctoral Fellowship to ZZ (18POST33960457), USDA/CRIS (51000-064-01S) to Y.X., the MCDB/BSCRC Microscopy Core, and the TCGB Technology Center for Genomics and Bioinformatics (supported by grant no. P30 CA016042). A.A. was supported by the National Institute for Mental Health (R00 MH106649 and R01 MH119089), the Brain and Behavior Research Foundation NARSAD Grant # 22663, and the Hellman Foundation. F.M.C.V.R. was supported with FAPESP grants #2015/23092-3 and #2017/08668-1 and a NARSAD grant #27654. We thank Dr. E. Gracheva and M. Massa for critical feedback on the manuscript and Dr. N. Rance for sharing methods to measure tail temperature in mice.

## Author contributions

S.M.C. and Z.Z. conceived of and designed the studies, analyzed the data, and wrote the manuscript with input from all authors. F.M.C.V.R., S.P., Z.Z., and A.A. designed, performed, and analyzed the fiber photometry studies. Y.H. and Y.X. designed, performed, and analyzed the electrophysiology experiments. Y.H. and J.E.V. acquired and analyzed the patch-seq data. J.R.D. and N.S. analyzed the thermal imaging data. Z.Z. and O.A. acquired and analyzed the electrocardiography data. M.S., Z.Z., and M.L. designed, performed, and analyzed the indirect calorimetry experiments. I.N., S.A., Z.Z., J.E.V., and K.P. designed, performed, and analyzed electroencephalography data. Z.Z., J.W.P., J.R.D., and M.G.M. performed chemogenetic and ablation experiments and analyzed the data.

## Competing interests

The authors declare no competing interests.
