## [Peer Review File · Nature Communications]

REVIEWERS' COMMENTS -

Reviewer #1 (Remarks to the Author):

The authors have done a nice job addressing the suggestions and comments. I have a few minor suggestions.

1. Since the original review, I have looked at POA scRNA-seq datasets and ESRa is really quite widely distributed in many neuronal clusters. This point could be made more explicit in the ms.
2. In light of Hrvatin and Takahashi, the statement (lines 29-32) "Despite recent advances in our understanding of thermoregulation, the precise neurons and neural circuitry that coordinate these profound thermoregulatory and metabolic changes are largely unknown. " is not quite correct.
3. 90 "Overall, ERa immunoreactivity in the MPA was higher in females than in males (Fig.1a and Extended Data Fig. 1a)." These are anecdotal and not quantitated.
4. Nomenclature of POA regions, including authors' response to reviewer 2
Correa et al. use MPA to indicate the very rostral part of ventral medial POA, which is referred to as MPA in the Franklin and Paxinos mouse brain atlas (please correct the reference, spelling out Paxinos). Despite that, some researchers have, probably rightfully so, dissented with the atlas' interpretation and refer to this part of the POA as the median preoptic nucleus (MnPO). Abbott and Saper, 2017: "The median preoptic nucleus in mice, as in rats, is an inverted Y-shaped structure that is draped over the rostral end of the third ventricle. The two arms of the Y are located along either side of the third ventricle as it opens and enclose the organum vasculosum of the lamina terminalis (OVLT), which is located along the rostral tip of the ventricle."
5. 210 I learned a lot from the other reviewer about the Boulant Q10 > 2 method for assigning 'heat sensitive'. It is interesting that whole body metabolic rate Q10 is closer to 2.8-3 (White (2003) PNAS 100:4046), so one might have expected that an individual cell would be disproportionately sensitive, ie say Q10 > 3 (not 2). It would be instructive to use a Q10 > 3 to see if that is more discriminating in your data.
6. 342 the statement "Our results suggest that ERa+ MPA neurons are essential for maintaining torpor in both female and male mice," is too strong. The ablation experiments show quantitative effects on torpor; not the loss of torpor, needed to say "essential".
7. 309 "torpid state lasted from 20 h to multiple days, far outlasting the pharmacokinetic duration of CNO (1-3 h82)" I would assume that CNO metabolism is greatly slowed by the hypometabolic state in torpor, so the PK duration should be much longer. Is there any data on this?
8. 329 while many, many compounds can induce torpor like states, including adenosine A1 agonists, knock out of adenosine A1 receptor does not affect entry into torpor.
10.1016/j.neuropharm.2016.11.026
9. there are a number of places where the phrasing seems a little off. While each is minor or trivial, the overall impression is that the ms misses on nuances of the physiology. It would be good to convey more clearly the message that torpor is a highly regulated and very desirable, beneficial aspect of physiology.
Examples by line number:
47 "modulate heat production, heat dissipation, and basal metabolic rate" isn't BMR is a subset of heat production?

48 "effectors are energetically demanding and may be compromised when energy" why 'compromised' as opposed to 'down-regulated' or some other way of conveying that this is a highly regulated and beneficial thing, not a loss of control

50 "in which body temperature, physical activity, metabolism and reproduction are usually dramatically reduced" why the word usually?

53 "neuronal populations within the POA that respond to temperature changes" how about 'local' temperature changes.

146 "The expression of hypothermia is highly associated with energy state." Hypothermia in the mouse is achieved by turning off BAT; this is a very indirect sentence.

And "Accordingly, body weight was significantly associated with the minimum core temperature reached and with duration of the hypothermia following CNO administration" not sure what the accordingly refers to, but the smaller body weight means less fat and thus less energy stores and possibly also a contribution from increased heat loss due to relatively higher surface area: volume.

148 "note, even in a thermoneutral environment (30°C), when thermogenesis to defend body temperature is diminished" there is recent work on what thermoneutrality is, see 10.1016/j.celrep.2020.03.065

Reviewer #3 (Remarks to the Author):

The authors have made extensive revisions that address the concerns of each of the reviewers. This is an excellent paper making significant contributions to our understanding of torpor mechanisms. I can only hope that they proceed to fulfill their suggestions of future experimentation into this exciting research.

Line 385 should read "appear to include...".

Indeed, the extent to which these neurons are "temperature sensitive" is not particularly relevant to the overall findings. The authors' concern seems to have been that the hypothermia of torpor could arise from the activation of warm-sensitive neurons which would then elicit behavioral and autonomic responses to reduce T_{core}. However, since the thermal stimulus for mouse torpor is a low ambient temperature, it is hard to understand how the authors imagine that the activity of warm-sensitive neurons would be increased during torpor induction, thereby driving a hypothermia.

Reviewer #4 (Remarks to the Author):

This study provides interesting information about the role of ER α neurons in the medial preoptic area (MPA) as drivers of hypothermia and hypometabolism in mice. Activation of these neurons leads to torpor, as demonstrated by the authors by chemogenetics approaches. A deep metabolic phenotyping is also provided. Notably, the mechanism shows sex-dependent differences.

This is an excellent study that focusses on a relevant topic. The experimental approach is sound, and it adds interesting (and physiologically relevant) information about the role of estrogen sensitive MPA neurons in the regulation of metabolism in response to energy deficiency. My only main suggestions at this stage are minor and mainly formal:

1. The neuronal pathway modulating the effect of ER α MPA neurons is demonstrated to depend on the ARC and DHM (but not the VMH). While these experiments offer interesting preliminary data, the evidence is still too initial. Explaining this limitation and a deeper discussion of the potential ARC and DMH differences would be of interest.
2. Discussion section is too dense (5.5 pages) and it can be clearly shorten

3. The number of references (119) is also excessive.

We thank the referees again for their careful reading of our manuscript and thoughtful suggestions, especially in this unusual situation. We have carefully addressed each of the referee comments in the revised manuscript, as outlined below and highlighted in blue. Thanks to your valuable input, we believe that the revised manuscript is greatly improved.

Referee #1 (Remarks to the Author):

The authors have done a nice job addressing the suggestions and comments. I have a few minor suggestions.

1. Since the original review, I have looked at POA scRNA-seq datasets and *Esr1* is really quite widely distributed in many neuronal clusters. This point could be made more explicit in the ms.

We agree that *Esr1* transcripts are detected in many neuronal clusters of the MPA, especially these involved in thermoregulation and metabolic processes such as *Adcyap1*. We have added a sentence to explicitly state this point in lines 71-72: "Single cell RNA profiling of the POA has revealed that *Esr1* transcripts are expressed within many neuronal clusters that control body temperature and metabolism"¹⁴.."

2. In light of Hrvatin and Takahashi, the statement (lines 29-32) "Despite recent advances in our understanding of thermoregulation, the precise neurons and neural circuitry that coordinate these profound thermoregulatory and metabolic changes are largely unknown." is not quite correct.

We have changed the statement in lines 29-32 to:

Recent advances have revealed the medial preoptic area (MPA) of the hypothalamus as a key site for the regulation of torpor in mice. The MPA is estrogen-sensitive and estrogens also have potent effects on both temperature and metabolism.

3. 90 "Overall, ER α immunoreactivity in the MPA was higher in females than in males (Fig.1a and Extended Data Fig. 1a)." These are anecdotal and not quantitated.

We have added quantification to support this statement (Supplementary Fig. 1b).

4. Nomenclature of POA regions, including authors' response to reviewer 2
Correa et al. use MPA to indicate the very rostral part of ventral medial POA, which is referred to as MPA in the Franklin and Paxinos mouse brain atlas (please correct the reference, spelling out Paxinos). Despite that, some researchers have, probably rightfully so, dissented with the atlas' interpretation and refer to this part of the POA as the median preoptic nucleus (MnPO). Abbott and Saper, 2017: "The median preoptic nucleus in mice, as in rats, is an inverted Y-shaped structure that is draped over the rostral end of the third ventricle. The two arms of the Y are located along either side of the third ventricle as it opens and enclose the organum vasculosum of the lamina terminalis (OVLT), which is located along the rostral tip of the ventricle."

We thank the referee for clarifying the point made by reviewer 2. We have adjusted the outline of the MnPO in Fig. 1a,b and Supplementary Fig.1a,c. to drape along and follow the third ventricle. However, it does not reach the most ventral aspect, to maintain accordance with the atlas and previous work on NK3R neurons and thermoregulation.

We also have spelled out Paxinos in the reference. Thank you for catching that.

5. 210 I learned a lot from the other reviewer about the Boulant $Q_{10} > 2$ method for assigning 'heat sensitive'. It is interesting that whole body metabolic rate Q_{10} is closer to 2.8-3 (White (2003) PNAS 100:4046), so one might have expected that an individual cell would be disproportionately sensitive, ie say $Q_{10} > 3$ (not 2). It would be instructive to use a $Q_{10} > 3$ to see if that is more discriminating in your data.

This is an interesting comparison. A Q_{10} of 3 was used based on the best fit of their regression model and Q_{10} effect only accounts for less than 15% of the correction in (White (2003) PNAS 100:4046). Nevertheless, it is interesting to try a cut-off of $Q_{10} > 3$. A detailed comparison is listed in the table below. As expected, fewer neurons are classified as warmth-responsive with the more stringent cut-off. We did not include this to the revised manuscript due to many unvalidated assumptions (e.g., assuming the whole body as a sum of individual cell metabolism and the same Q_{10} applies in vivo and in vitro). Additionally, a more permissive cut-off is more conservative in this case, as our goal was to exclude the alternative hypothesis that the neurons are only involved in thermoregulation.

POA regions	Q10>2		Q10>3	
	WR/Total	%of WR	WR/Total	%of WR
MPN	41/73	56.2%	38/73	52.1%
rMPA	20/35	57.1%	17/35	48.6%
VLPO	5/36	13.9%	5/36	13.9%
RLPO	10/35	28.6%	8/35	22.9%
MPA(MPN&rMPA)	61/108	56.5%	55/108	50.9%

6. 342 the statement "Our results suggest that ER α + MPA neurons are essential for maintaining torpor in both female and male mice," is too strong. The ablation experiments show quantitative effects on torpor; not the loss of torpor, needed to say "essential".

We have changed the statement (line 335) into "...are **involved in** maintaining torpor ..."

7. 309 "torpid state lasted from 20 h to multiple days, far outlasting the pharmacokinetic duration of CNO (1-3 h²)" I would assume that CNO metabolism is greatly slowed by the hypometabolic state in torpor, so the PK duration should be much longer. Is there any data on this?

This is a great perspective. We do not have data to address this hypothesis. However, with a correction of $Q_{10}=3$ as the reviewer mentioned above, the PK duration of CNO could be 3 times slower over a 10 °C decrease of T_{core} . This would result in a PK duration of 3-12h. Although this PK duration cannot fully explain the duration of torpor and does not alter the conclusion, we added this thoughtful point to the discussion. Lines 305-306: "...far outlasting the pharmacokinetic duration of CNO (1-3 h⁷¹, although the duration could be longer if torpor also reduces CNO metabolism)."

8. 329 while many, many compounds can induce torpor like states, including adenosine A1 agonists, knock out of adenosine A1 receptor does not affect entry into torpor. 10.1016/j.neuropharm.2016.11.026

We thank the reviewer for pointing out this essential evidence. We have changed the statement in line 326 into: "...a torpor-like state in rats but ablation of the adenosine A1 receptor does not affect entrance into torpor."

9. there are a number of places where the phrasing seems a little off. While each is minor or trivial, the overall impression is that the ms misses on nuances of the physiology. It would be good to convey more clearly the message that torpor is a highly regulated and very desirable, beneficial aspect of physiology.

We thank the review for these thoughtful suggestions in improving the readability of the manuscript. We have made the following changes according to your comments.

Examples by line number:

47 “modulate heat production, heat dissipation, and basal metabolic rate” isn’t BMR is a subset of heat production?

Line 47 was changed to "...modulate **facultative thermogenesis**, heat dissipation,...".

48 “effectors are energetically demanding and may be compromised when energy” why ‘compromised’ as opposed to ‘down-regulated’ or some other way of conveying that this is a highly regulated and beneficial thing, not a loss of control

Line 48 was changed to “effectors are energetically demanding and may be **down-regulated** when energy...”

50 “in which body temperature, physical activity, metabolism and reproduction are usually dramatically reduced” why the word usually?

We deleted the word “usually” in the manuscript.

53 “neuronal populations within the POA that respond to temperature changes” how about ‘local’ temperature changes.

Line 53 was changed into “..neuronal populations within the POA that respond to **local** temperature changes...”

146 “The expression of hypothermia is highly associated with energy state.” Hypothermia is in the mouse is achieved by turning off BAT; this is a very indirect sentence.

And “Accordingly, body weight was significantly associated with the minimum core temperature reached and with duration of the hypothermia following CNO administration” not sure what the accordingly refers to, but the smaller body weight means less fat and thus less energy stores and possibly also a contribution from increased heat loss due to relatively higher surface area: volume.

We apologize for the confusion. We would like to make the point that similar to “natural” torpor (induced by fasting in mice), which body weight and energy state are correlated with torpor expression, we observed similar association. We agree that this association may partly due to an increased heat loss from higher surface to volume ratio.

We have changed the statement in lines 143-144 as “The expression of torpor has been previously associated with body weight⁵⁰⁻⁵². We also find that body weight was significantly correlated with the minimum core temperature reached and with duration of the hypothermia following CNO administration (Supplementary Fig. 2b).”

148 “note, even in a thermoneutral environment (30oC), when thermogenesis to defend body temperature is diminished” there is recent work on what thermoneutrality is, see 10.1016/j.celrep.2020.03.065

We thank the reviewer for suggesting this more current conceptual framework. We have changed the sentence in line 146 as " ... even in warmer conditions (ambient temperature of 30°C) that might diminish thermogenesis,..."

Referee #3 (Remarks to the Author):

The authors have made extensive revisions that address the concerns of each of the reviewers. This is an excellent paper making significant contributions to our understanding of torpor mechanisms. I can only hope that they proceed to fulfill their suggestions of future experimentation into this exciting research.

We thank the referee again for their valuable input.

Line 385 should read “appear to include...”.

We have corrected the error in the revised manuscript.

Indeed, the extent to which these neurons are “temperature sensitive” is not particularly relevant to the overall findings. The authors’ concern seems to have been that the hypothermia of torpor could arise from the activation of warm-sensitive neurons which would then elicit behavioral and autonomic responses to reduce T_{core}. However, since the thermal stimulus for mouse torpor is a low ambient temperature, it is hard to understand how the authors imagine that the activity of warm-sensitive neurons would be increased during torpor induction, thereby driving a hypothermia.

We appreciate that the question of temperature sensitivity is not a central concern. To clarify, we were not proposing that warmth-responsive neurons would be drivers of torpor. We only felt that it was important to exclude the possibility that warm- responsive neurons were causing hypothermia without torpor. To exclude this hypothesis, we took two approaches: 1) look at the timing in the changes in metabolism and body temperature to confirm that hypothermia does not precede hypometabolism and 2) ask if the ER α MPA neurons are (uniformly) warmth responsive.

Referee #4 (Remarks to the Author):

This study provides interesting information about the role of ER α neurons in the medial preoptic area (MPA) as drivers of hypothermia and hypometabolism in mice. Activation of these neurons leads to torpor, as demonstrated by the authors by chemogenetics approaches. A deep metabolic phenotyping is also provided. Notably, the mechanism shows sex-dependent differences.

This is an excellent study that focusses on a relevant topic. The experimental approach is sound, and it adds interesting (and physiologically relevant) information about the role of estrogen sensitive MPA neurons in the regulation of metabolism in response to energy deficiency. My only main suggestions at this stage are minor and mainly formal:

We thank the referee for their willingness to step in as a reviewer and their positive evaluation of the manuscript.

1. The neuronal pathway modulating the effect of ER α MPA neurons is demonstrated to depend on the ARC and DHM (but not the VMH). While these experiments offer interesting preliminary data, the evidence is still too initial. Explaining this limitation and a deeper discussion of the potential ARC and DMH efferences would be of interest.

We agree that the results of delivering CNO to ARC and DMH are preliminary and require more selective circuit mapping and manipulations. We now state this explicitly (lines 397-400): “Although more selective circuit mapping and manipulations are required to fully understand the circuits that control torpor, these studies provide additional evidence of a potent role for ER α + MPA neurons in the regulation of torpor.” Additionally, we removed the second half of the sentence, which used to say “...and suggest that they may coordinate the rapid drop in temperature at least partially through projections to the ARC and DMH.” Removing this statement limits the interpretation of the data to the role of the neurons rather than the role of the projections, which we agree is the prudent interpretation of our results.

We also modified line 411 “...whereas our preliminary data suggest that ER α + MPA neurons may exert their effects through projections to the ARC and partially through the DMH.”

2. Discussion section is too dense (5.5 pages) and it can be clearly shorten

We agree that the discussion is quite long and have streamlined the text to some degree, reducing the length to 4.5 pages. We could not cut anymore because we wanted to avoid overlooking the various points raised in this and the previous review.

3. The number of references (119) is also excessive.

We have removed some of the references, reducing the total number to 103 references.